# A Highly Sensitive Next-Generation Sequencing-Based Genotyping Platform for *EGFR* Mutations in Plasma from Non-Small Cell Lung Cancer Patients

**DOI:** 10.3390/cancers12123579

**Published:** 2020-11-30

**Authors:** Jung-Young Shin, Jeong-Oh Kim, Mi-Ran Lee, Seo Ree Kim, Kyongmin Sarah Beck, Jin Hyoung Kang

**Affiliations:** 1Laboratory of Medical Oncology, Cancer Research Institute, College of Medicine, The Catholic University of Korea, Seoul 06591, Korea; bearjy@catholic.ac.kr (J.-Y.S.); kjo9713@catholic.ac.kr (J.-O.K.); miran13@catholic.ac.kr (M.-R.L.); 2Department of Medical Oncology, Seoul St. Mary’s Hospital, The Catholic University of Korea, Seoul 06591, Korea; 21300424@cmcnu.or.kr; 3Department of Radiology, Seoul St. Mary’s Hospital, The Catholic University of Korea, Seoul 06591, Korea; sallahbar@catholic.ac.kr

**Keywords:** *EGFR* mutation, EGFR-TKI, cfDNA, NGS, liquid biopsy, digital enrichment

## Abstract

**Simple Summary:**

In this study, Sel-Cap^TM^, a next-generation sequencing (NGS)-based genotyping platform, showed high sensitivity for detection of *epidermal growth factor receptor* (*EGFR*) gene mutations in plasma samples collected from 185 patients with non-small cell lung cancer (NSCLC). In the early-stage NSCLC, Sel-Cap liquid biopsy was able to detect more than half the *EGFR* mutations, which were detected in tumor tissue (sensitivity: 50% and 78% for Ex19del and L858R respectively, with tumor results as the references), while the conventional NGS could not detect any. Sel-Cap liquid biopsy was particularly sensitive for resistant mutation T790M (sensitivity: 88%). In addition, we conducted a retrospective study to monitor T790M using Sel-Cap in 34 patients who progressed on first-line tyrosine kinase inhibitors (EGFR-TKIs). The study suggested that the first appearance of T790M in plasma, ranging from at treatment baseline to over three years post-EGFR-TKI initiation, may be useful for prediction of disease progression (around 5 months in advance).

**Abstract:**

Sel-Cap^TM^, a digital enrichment next-generation sequencing (NGS)-based cancer panel, was assessed for detection of *epidermal growth factor receptor* (*EGFR*) gene mutations in plasma for non-small cell lung cancer (NSCLC), and for application in monitoring *EGFR* resistance mutation T790M in plasma following first-line EGFR-tyrosine kinase inhibitor (EGFR-TKI) treatment. Using Sel-Cap, we genotyped plasma samples collected from 185 patients for mutations Ex19del, L858R, and T790M, and compared results to those of PNAclamp^TM^ tumor biopsy (reference method, a peptide nucleic acid-mediated polymerase chain reaction clamping) and two other NGS liquid biopsies. Over two-thirds of activating mutations (Ex19del and L858R), previously confirmed by PNAclamp, were detected by Sel-Cap, which is 4–5 times more sensitive than NGS liquid biopsy. Sel-Cap showed particularly high sensitivity for T790M (88%) and for early-stage plasma samples. The relationship between initial T790M detection in plasma and progression-free survival (PFS) following first-line EGFR-TKIs was evaluated in 34 patients. Patients with T790M detected at treatment initiation (±3 months) had significantly shorter PFS than patients where T790M was first detected >3 months post treatment initiation (median PFS: 5.9 vs. 26.5 months; *p* < 0.0001). However, time from T790M detection to disease progression was not significantly different between the two groups (median around 5 months). In conclusion, Sel-Cap is a highly sensitive platform for *EGFR* mutations in plasma, and the timing of the first appearance of T790M in plasma, determined via highly sensitive liquid biopsies, may be useful for prediction of disease progression of NSCLC, around 5 months in advance.

## 1. Introduction

Nearly one in every five cancer deaths worldwide is caused by lung cancer (World Health Organization Report on Cancer, 2020, https://apps.who.int/iris/rest/bitstreams/1267643/retrieve). Non-small cell lung cancer (NSCLC) makes up the vast majority of all lung cancer cases, and approximately three-quarters of NSCLC patients are diagnosed at advanced-stage. Currently, the first-line systemic treatment for advanced-stage NSCLC is targeted therapy for those who bear driver oncogene mutations in tumor, for example, epidermal growth factor receptor (EGFR) tyrosine kinase inhibitors (TKIs) for patients with drug-activating mutations in the *EGFR* gene [1].

Exon 19 deletion (Ex19del) and exon 21 L858R are the most frequent *EGFR*-activating mutations, and the secondary gatekeeper mutation T790M, which can result from long-term exposure to first-line EGFR-TKIs, is one of the primary causes for acquired EGFR-TKI resistance [2]. During re-biopsy, T790M is found in over half of the tumor samples taken from EGFR-TKI-resistant patients [3]; however, tumor re-biopsy is usually performed post tumor relapse, and is often not feasible in clinical situations such as those involving patients in poor physical condition and/or with hardly accessible target lesions.

Cell-free DNA (cfDNA) refers to all nucleic acid fragments circulating in blood; in cancer patients, 0.01% to 90% cfDNA may consist of tumor-derived DNA [4]. Several detection techniques for *EGFR* mutations in plasma-derived cfDNA have been developed as non-invasive alternatives to tumor *EGFR* genotyping [5], such as cobas^®^ EGFR mutation test v2, BEAMing-PCR (BEAM refers to Beads, Emulsions, Amplification and Magnetics) PCR [6], ARMS-PCR (ARMS refers to Amplification Refractory Mutation System), and ddPCR^TM^ (dd refers to Droplet Digital). These techniques are characterized by quantitative results and a short turnaround time, but are limited to pre-defined mutations, and the sensitivity needs improvement, particularly in early-stage disease [7].

Sel-Cap lung cancer panel (hereinafter referred to as Sel-Cap) is a next-generation sequencing (NGS)-based oncogene genotyping platform, equipped with a pre-sequencing mutation-enrichment feature [8] (limit of detection: 0.01–0.05%, limit of detection is defined as the percentage of mutation copies that must be present in the specimen for a mutation to be identified). The primary objective of this study was to evaluate Sel-Cap’s capacity for detecting *EGFR* mutations (Ex19del, L858R, and T790M) in plasma-derived cfDNA, by comparing it to other commonly used genotyping platforms such as peptide nucleic acid clamping (PNAclamp; currently, the most popular platform in Korea) in tumor, as well as conventional NGS (which is a non-commercialized mutation panel based on the commonly used NGS technique) and an NGS-based cancer panel in plasma. In addition, to explore Sel-Cap’s potential application in monitoring for *EGFR* resistance mutation T790M in plasma to predict resistance to first-line EGFR-TKI treatments, a retrospective longitudinal study was carried out in patients who exhibited this resistance.

## 2. Materials and Methods

### 2.1. Study Population

Plasma samples used in this study were collected from patients histologically diagnosed with NSCLC (adenocarcinoma) between January 2011 and January 2019 in Seoul St. Mary’s Hospital. All samples were stored by Seoul St. Mary’s Hospital biobank. Before sample collection for the biobank, all the patients provided a written informed consent for the possible use of their samples in the future research.

In this study, only samples which had been previously genotyped by PNAclamp *EGFR* mutation detection kit ver.2 (PNAC-3002, Panagene, Daejeon, Korea) were included. This study was approved by the institutional review board (IRB) in Seoul St. Mary’s Hospital (No. KC17TNSI0184), and was performed in accordance with the national laws, regulations, and good clinical practice (GCP) guidelines for patient data protection.

### 2.2. Sample Preparation

To prepare plasma samples, patients’ blood was drawn into ethylenediaminetetraacetic acid (EDTA) tubes, and was immediately centrifuged at 1200× *g* at 4 °C for 15 min. The supernatant was then transferred to 1.5 mL sterile Eppendorf tubes, and centrifuged at 13,000× *g* at 4 °C for 10 min [9]. The separated plasma was stored at −80 °C until use.

cfDNA was extracted from plasma samples using a DNeasy Blood and Tissue Kit (Qiagen, Hilden, Germany) according to the manufacturer’s instructions. The concentration of cfDNA was quantified using a Qubit^TM^ dsDNA HS Assay Kit (ds refers to double strand, HS refers to High-Sensitivity) with Qubit^TM^ 2.0 Fluorometer (Life Technologies, Carlsbad, CA, USA). The purity of cfDNA was evaluated with a NanoDrop ND-1000 Spectrophotometer (Thermo Fisher Scientific, Waltham, MA, USA), and only samples with an A260/A280 ratio of 1.8–2.0 passed quality control. The cfDNA was then end-repaired and size selection was performed, followed by adenylation of the 3′ end [10].

### 2.3. EGFR Mutation Detection Platforms

In this study, four different platforms were used for *EGFR* mutation detection, and all procedures were carried out in accordance with the manufacturer’s instructions. PNAclamp, the standard diagnostic method for *EGFR* tumor biopsy in the hospital, was used as a reference, and the methodology was described in detail, previously [11]. The limit of detection of PNAclamp was determined to be >0.1%. In addition, a conventional NGS panel (Ion AmpliSeq™ Cancer Hotspot Panel, Life Technologies, Carlsbad, CA, USA), a 30-gene NGS lung cancer panel (Theragen, Suwon, Korea), and a Sel-Cap lung cancer panel (SeaSun Biomaterials, Daejeon, Korea) were used for comparison in plasma samples in two separate studies. The mutation detection cut-off value was 0.1% for Sel-Cap, and 1% for the conventional NGS and 2% for the 30-gene NGS lung cancer panel. The cut-off value was determined to be the average minimum variant allelic frequency that could be reliably detected (variant allelic frequency: percentage of mutant reads over total reads at one locus). Mutations in exon 18–20 were genotyped (Exon 18: L718Q, G719X; Exon 19: deletion; Exon 20: insertion, T790M, and S768I; Exon 21: L844V, L858R, L861Q).

### 2.4. Sel-Cap Mutation Enrichment PCR

In enrichment polymerase chain reaction (PCR), wild-type-specific blockers were used to preferentially hybridize with wild-type alleles and reduce the background wild-type sequence amplification, which therefore resulted in enrichment of mutant PCR fragments. The assay used 30 ng of cfDNA and was performed according to the manufacturer’s protocol (SeaSun Biomaterials, Daejeon, Korea). In addition, to improve the performance of NGS, nonspecific PCR products (mainly primer dimers) were removed by Agencourt AMPure XP beads (Beckman Coulter, Vienna, Austria) using a 1:1 DNA-to-bead ratio.

### 2.5. NGS Library Preparation

Sequencing library preparation PCR was performed using the following: 2 μL of purified PCR product from mutation enrichment PCR amplification as the template, *EGFR* Insight 2× Seq Lib Pep Premix (SeaSun Biomaterials, Daejeon, Korea), and barcoded primer pairs. For the library preparation PCR, multiple indexing adapters were ligated to the ends of DNA fragments, and DNA fragments with specific adapters were amplified. Any unwanted short fragments were removed with Agencourt AMPure XP beads (Beckman Coulter, Brea, CA, USA) using a 1:1 DNA-to-bead ratio. The insert size of the library was detected on an Agilent 2100 Bioanalyzer (Agilent Technologies, Inc., Santa Clara, CA, USA), and effective concentration of the library was accurately quantified using a Qubit^TM^ 2.0 Fluorometer (Life Technologies, Carlsbad, CA, USA).

### 2.6. Monitoring EGFR T790M in Plasma for EGFR-TKI Treatment

To explore the application of Sel-Cap in monitoring plasma *EGFR* resistance mutation T790M to predict resistance to first-line EGFR-TKIs, a retrospective inspection was conducted on the serial plasma samples collected from patients with advanced disease who (1) were treated with first/second-generation EGFR-TKIs (gefitinib, erlotinib, or afatinib) and (2) had already developed disease progression (PD). Progression-free survival (PFS) was defined as the interval between EGFR-TKI initiation and PD. Tumor response was assessed by imaging techniques (such as computed tomography and magnetic resonance imaging) and determined based on the Response Evaluation Criteria in Solid Tumors (RECIST) version 1.1 [12].

### 2.7. Statistical Analyses

The diagnostic performance of liquid biopsies was evaluated based on sensitivity, specificity, accuracy, and Kappa coefficient, with the PNAclamp tumor biopsy serving as the reference. Sensitivity was calculated as the percentage of positive diagnoses (*EGFR* mutations) by test platform vs. by reference platform. Specificity was calculated as the percentage of negative diagnoses (*EGFR* wild-type) by test platform vs. by reference platform. Accuracy was calculated as the percentage of positive plus negative diagnoses by test platform vs. by reference platform. A Kappa coefficient, a statistical measure used to assess agreement between platforms, between 0.6 and 0.8 is generally regarded as “substantial agreement”, while a Kappa coefficient over 0.8 is generally regarded as “almost perfect agreement”. Statistical analyses were performed using the SPSS (version 22.0) program (IBM Corporation, Armonk, NY, USA).

## 3. Results

### 3.1. Study Population

The CONSORT flow diagram is presented in Figure 1. We identified 250 eligible patients whose tumor samples were available and previously genotyped for *EGFR* mutations by PNAclamp, and 185 of those patients had plasma samples available. The median age of the 185 patients at diagnosis was 64 years old, and the ratio of male to female was about 4 to 5. Nearly half of the patients (57.3%) were in TNM stage III/IV (TNM: a globally standardized cancer staging system, T: primary tumor, N: regional lymph node, M: distant metastasis).

Three separate studies were conducted to evaluate Sel-Cap liquid biopsy, using plasma samples collected from 185 different patients. In the first study, plasma samples from 61 patients were tested for Ex19del and L858R by both Sel-Cap and conventional NGS (T790M was not tested because these patients’ PNAclamp tumor biopsies did not include T790M); in the second study, plasma samples from all 185 patients were genotyped for Ex19del, L858R, and T790M by Sel-Cap, and in the third study, plasma samples were collected from 21 patients after they had developed resistance to first-line EGFR-TKIs and genotyped using both Sel-Cap and the NGS cancer panel. Finally, for the retrospective longitudinal T790M monitoring study, out of the patients with T790M-positive plasmas who progressed on first-line EGFR-TKIs, 34 eligible patients were identified and divided into two groups (early T790M detection and late T790M detection, based on the first time T790M was detected in plasma). Patients in the late T790M detection group all had serial plasma samples taken every 3–6 months along with a tumor response evaluation by imaging.

### 3.2. Sel-Cap Showed High Sensitivity for EGFR Mutations in Plasma

In the first study, the diagnostic performance of Sel-Cap liquid biopsy and a conventional NGS liquid biopsy in 61 patients, looking at Ex19del and L858R, is presented in Table 1 (with the PNAclamp tumor biopsy as the reference). The sensitivity of Sel-Cap liquid biopsy (75% for Ex19del and 65% for L858R) is 4–5 times higher than that of NGS (17% for Ex19del and 13% for L858R). When the results were stratified by disease stage, NGS showed a low sensitivity for Ex19del and L858R (36% and 22%, respectively) in advanced-stage disease, and was unable to detect any mutations in early-stage disease, while Sel-Cap showed good sensitivity in advanced-stage (72% and 78% for Ex19del and L858R, respectively) and early-stage (78% and 50%, respectively) plasma samples (Figure 2).

Although it is not the primary objective of this study, Sel-Cap was also evaluated for tumor biopsy and showed almost perfect agreement (Kappa coefficient = 1.00 and 0.96 for Ex19del and L858R, respectively) with PNAclamp tumor biopsy (Appendix A). 

In the second study, the concordance between genotyping results from Sel-Cap liquid biopsy and PNAclamp tumor biopsy in 185 patients is presented in Table 2. The sensitivity for Ex19del, L858R, and T790M are 72%, 67%, and 88%, respectively. The concordance data for T790M was only calculated in 85 (out of 185) patients whose tumor samples were tested for T790M by PNAclamp, and plasma and tumor samples were collected within an interval of less than 30 days, because if there is a large time interval between plasma and tumor sample collection, the sensitivity of Sel-Cap may have been overestimated due to the occurrence of acquired T790M mutation.

In the third study, plasma samples collected from first-line EGFR-TKI-resistant patients (n = 21) were genotyped by Sel-Cap and NGS cancer panel (Table 3). Sel-Cap discovered three times more T790M-positive plasma samples (n = 12) than NGS cancer panel (n = 4), and two plasma samples were determined to be T790M-positive by both methods.

### 3.3. Timing of First T790M Detection in Plasma is Critical for PFS of First-Line EGFR-TKIs

Forty-eight patients who developed drug resistance to first-line EGFR-TKIs (gefitinib, erlotinib, or afatinib) and had T790M-positive plasma samples were identified (Figure 1), and 26 patients had single-point plasma samples while the rest of the 22 patients had serial plasma samples which were collected every 3 months along with a tumor response evaluation by imaging techniques (the actual sampling time varied between patients). In the study to clarify the relationship between the timing of first T790M detection and PFS, single-point T790M-positive plasma samples that did not clearly show whether the mutation was detected for the first time were excluded (n = 14). The clinical characteristics of included patients (n = 34) are shown in Appendix A.

The PFS following first-line EGFR-TKI treatment, and the intervals between the first T790M detection and PD, are shown in Figure 3. The PFS is significantly longer in patients where initial detection of T790M was >3 months after treatment initiation (late T790M detection), compared to patients where initial T790M detection was within ±3 months of treatment initiation (early T790M detection): the median PFS is 26.5 (range: 11.6–50.2) months vs. 5.9 (range: 1.2–24.1) months, respectively (Logrank test: hazard ratio [HR] = 4.2, 95% confidence interval [CI]: 1.7–10.6, *p* < 0.0001). In addition, the median interval between the first T790M detection and PD is 3.6 (range: 0–18.6) months for the late T790M detection group vs. 5.8 (range: 1.2–24.2) months for the early T790M detection group, which is not significantly different (*p* = 0.27). Acquired resistance to EGFR-TKI is clinically determined at least 6 months after treatment using diagnostic imaging, but T790M can be detected in plasma 2–3 months before diagnosis of acquired resistance; therefore, a cutoff at 3 months post-EGFR-TKI initiation was used in this study. The timeline plots of these patients are shown in Appendix A.

## 4. Discussion

In this study, we evaluated the Sel-Cap lung cancer panel, an NGS-based genotyping platform for the detection of *EGFR* mutations in patients with NSCLC, and we focused specifically on its diagnostic performance while serving as a liquid biopsy platform. To do so, Sel-Cap was compared to a standard tumor biopsy (PNAclamp) and two other liquid biopsy platforms (conventional NGS and NGS-based cancer panel) using plasma samples collected from 185 patients.

PNAclamp is currently the most popular diagnostic platform for *EGFR* mutations in Korea. In this study, PNAclamp tumor biopsy is used as the reference for all liquid biopsies. In the first study, Sel-Cap was compared to conventional NGS for detecting *EGFR*-activating mutations in 61 plasma samples. Sel-Cap showed 4–5 times higher sensitivity than NGS in plasma (75% vs. 17% for Ex19del, and 65% vs. 13% for L858R). The second study consisted of a larger sample size (n = 185) and was consistent with the first study: Sel-Cap detected over two-thirds of *EGFR*-activating mutations that were detected by PNAclamp in tumor (sensitivity: 72% for Ex19del and 67% for L858R). Sel-Cap showed very high sensitivity (88%) for *EGFR* resistance mutation T790M in plasma samples, and among the 85 plasma samples tested, Sel-Cap identified 14 more T790M-positive samples which the PNAclamp tumor biopsy was unable to detect.

Our recently published work evaluating a popular liquid biopsy cobas^®^ EGFR mutation test v2 (cobas v2) shows that its sensitivity for *EGFR* mutations is NSCLC stage-dependent [13]. In advanced-stage disease, the sensitivity of cobas v2 liquid biopsy is satisfactory for Ex19del (over 80%) but falls short in L858R and T790M (both are below 40%); furthermore, in early-stage disease, cobas v2 shows disappointingly low sensitivity [13]. In the current study, which looked at 61 plasma samples, the sensitivity of Sel-Cap liquid biopsy for both Ex19del and L858R was high, not only in advanced-stage disease (>70%), but it also showed much higher sensitivity than cobas v2 in early-stage disease (78% vs. 13% for Ex19del, and 50% vs. 0% for L858R). Although Sel-Cap and cobas v2 were not compared directly, both of the studies used an established tumor biopsy for reference (PNAclamp and cobas v2 tumor biopsy, respectively. Our previous study showed that the Kappa co-efficient of the two tumor biopsies was 0.82, indicating almost perfect agreement [13]); therefore, this indirect comparison is reasonably justifiable. In the future, the clinical significance of *EGFR* mutations detected in plasma at early-stage should be studied, and may be particularly valuable in clarifying the connection between *EGFR* mutations in plasma and the risk of tumor relapse, especially considering tumor *EGFR* mutation status-associated EGFR protein expression is a significant risk factor for tumor relapse in early-stage NSCLC [14,15]. Finally, in the 21 plasma samples collected from advanced-stage patients who developed drug resistance to first-line EGFR-TKIs, Sel-Cap detected three times more T790M-positive plasma samples than a different NGS-based cancer panel.

To the best of our knowledge, the sensitivity of Sel-Cap liquid biopsy is the highest among all liquid biopsy platforms on record for *EGFR* mutations in early-stage NSCLC, and Sel-Cap shows one of the highest sensitivities for overall disease stage, which is comparable to ddPCR, currently regarded as the most sensitive liquid biopsy to date [16,17]. The high sensitivity of Sel-Cap for liquid biopsy is attributed to its mutation-enrichment feature. NGS-based cancer panels usually require PCR amplification of regions of interest prior to sequencing, however, the amplification of low copy number mutations tends to be exponentially less efficient than that of the wild-type allele, often leading to selective negativity of the mutations (PCR bias) [18]. In order to solve the problem, Sel-Cap assay uses wild-type-specific blockers to suppress the amplification of wild-type alleles and thus can preferentially amplify mutant alleles. In addition to the high sensitivity, another clinically meaningful advantage of Sel-Cap liquid biopsy is that less plasma (0.7–0.8 mL) is required for each test compared to other platforms (2–10 mL).

For some of the patients in the study, T790M mutation was detected in plasma by Sel-Cap even at what is considered the baseline for first-line EGFR-TKIs (treatment initiation within ±3 months). Our study did not calculate the portion of T790M-positive patients among all EGFR-TKI-naïve patients, but previous studies estimate that T790M can be found in around 2% of *EGFR*-activating mutation-positive EGFR-TKI-naïve plasma samples [19] and frozen tumor samples [20]; however, the percentage can be much higher (>40%) in FFPE tumor samples, likely due to higher incidence of false positives [20]. This disadvantage (false positivity) of *EGFR* testing using FFPE tumor also suggests that if liquid biopsy sensitivity can rival that of tumor biopsy, it may become a good surrogate for tissue *EGFR* testing [21], since it usually shows low false positivity.

It is estimated that nearly half of first-line EGFR-TKI-treated patients may acquire the T790M mutation after long-term treatment [3], which is in line with the T790M-positive rate observed in the 21 post-PD plasma samples in our study. Before liquid biopsy platforms were available, the conventional T790M detection was conducted through tumor re-biopsy after disease progression, and the prognostic value of T790M diagnosis was very limited [22]. In recent years, an increasing number of studies on monitoring T790M in plasma have been conducted [17,19,23,24,25,26]. In our study, excluding those who were already T790M-positive at treatment baseline (early T790M detection group), the median time for the initial detection of T790M in plasma (in late T790M detection group) was 23.4 months (ranging from 5.5 to 38.2 months) post-EGFR-TKI initiation, which is similar to the results reported in a recently published monitoring study [23]. Interestingly, regardless of whether T790M is initially detected at baseline or after long-term treatment, the time to PD post-T790M detection was not significantly different (median time ~5 months). Several previous studies conducted similar longitudinal monitoring for T790M in NSCLC patients who had tumor progression on EGFR-TKI treatments [19,24]. In their studies, 40–70% of the patients showed initial T790M detection in plasma at the time of PD or after. For those whose T790M first appeared before PD diagnosis, the time between the detection and PD was typically 2–3 months. Our study shows that Sel-Cap can not only detect T790M before PD for the vast majority of patients (16/18 patients, according to the late T790M detection group), but it can also predict PD at least 2–3 months earlier than an ordinary liquid biopsy, and therefore could increase the time window for physicians to plan the next treatment and monitoring programs.

In addition, our study emphasized that monitoring programs should be individualized. For instance, because the patients with T790M detected at baseline tend to develop drug resistance to first-line EGFR-TKIs much faster than those who have T790M detected >3 months later, a monitoring program with more frequent intervals may be needed for patients with baseline T790M. On the other hand, in patients without baseline T790M, since the longitudinal monitoring period for initial T790M detection varies vastly, for those who continuously respond to first-line EGFR-TKIs, longer monitoring intervals between liquid biopsies can be considered to ease the financial burden of testing. More importantly, in future studies, the risk factors for the occurrence of T790M in plasma should be investigated and considered [27] so that monitoring of T790M by liquid biopsy may become more efficient. Previous studies found that T790M was more prevalent in metastatic tumors than primary tumors in NSCLC patients [28], and tumor size was positively and significantly correlated with cfDNA level in ovarian cancer [29] and melanoma [30]. Our team is currently looking for correlations between primary and/or metastatic tumor size, *EGFR* mutations in plasma, and disease progression.

This study has several limitations. First, in the study with 185 plasma samples (the third study), the sensitivity of Sel-Cap was not stratified by disease stage because this information was not known for all of the samples. Second, all the patients included in the retrospective monitoring study had at least one T790M-positive plasma sample and developed disease progression on first-line EGFR-TKI treatments, and the study did not include samples collected from those who might have missed diagnosis of T790M due to single sampling, or from those who were still responding to first-line EGFR-TKI treatments. A prospective study is needed to validate the clinical application of Sel-Cap for prediction of disease progression, compared to the standard imaging diagnostic techniques. Third, in this real-world study, the plasma sampling for the longitudinal monitoring of T790M did not strictly follow a schedule; in some patients, the actual T790M detection time in plasma could have been a few months earlier.

Sensitive and reliable genotyping platforms are the premise of successful lung cancer target therapy. Ideally, such platforms should also be non-invasive, fast, and affordable. So far, our study has generated encouraging results for the application of Sel-Cap, a highly sensitive digital enrichment NGS liquid biopsy for NSCLC. With the increasing discovery of other EGFR-TKI resistance mechanisms besides T790M (for example, *EGFR* mutation copy number [26] and c-met overexpression [31]) and with the increasing understanding of other factors influencing the detection of T790M in plasma-derived cfDNA, we may be able to predict the occurrence of drug-resistance more accurately and determine the optimal time to switch to the third-generation EGFR-TKIs. We foresee that regular monitoring for *EGFR* mutations with sensitive liquid biopsy platforms, such as Sel-Cap, may become the standard of practice in the future precision medicine for NSCLC. 

## 5. Conclusions

In this study conducted using plasma samples collected from 185 NSCLC patients, Sel-Cap, a digital enrichment NGS-based lung cancer panel, shows very high sensitivity for *EGFR* mutations in cfDNA, even in early-stage disease. The application of Sel-Cap liquid biopsy in a retrospective, longitudinal monitoring study suggests that highly sensitive platforms for *EGFR* resistance mutation T790M in plasma may allow for prediction of disease progression around 5 months in advance, unlike tumor re-biopsy or less sensitive liquid biopsy, where the T790M is often detected after disease progression.

## Figures and Tables

**Figure 1 cancers-12-03579-f001:**
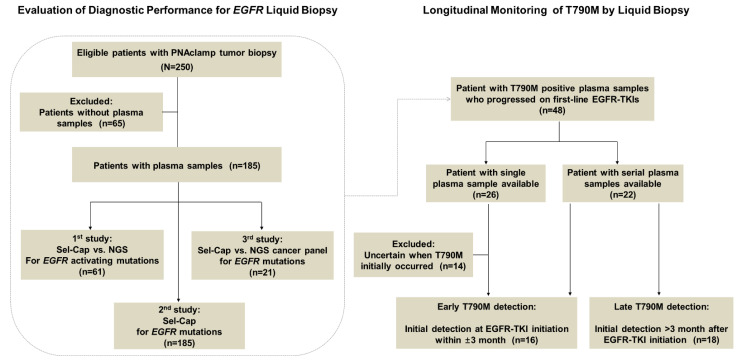
CONSORT flow diagram for the present study. NGS, next-generation sequencing; EGFR-TKI, epidermal growth factor receptor-tyrosine kinase inhibitor.

**Figure 2 cancers-12-03579-f002:**
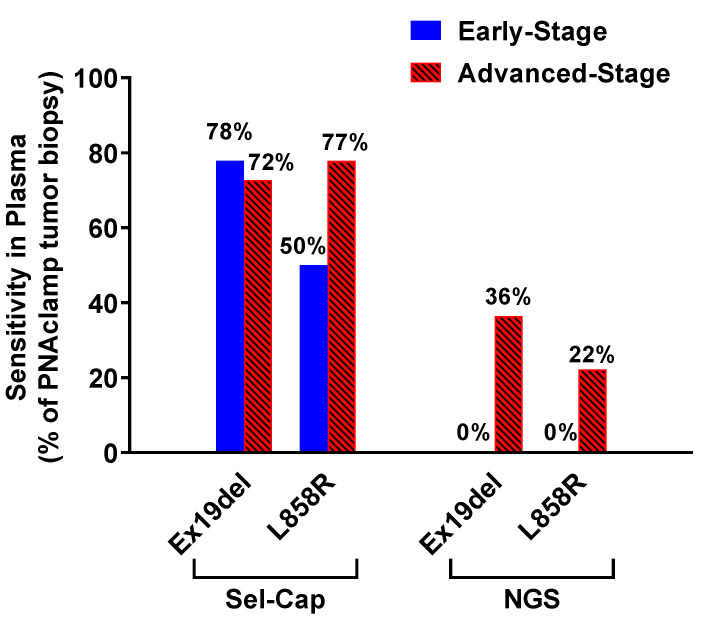
Comparison of sensitivity between Sel-Cap and conventional NGS liquid biopsies for two *EGFR*-activating mutations in NSCLC patients (n = 61, Table 1), stratified by disease stage (n = 30 and 31 for early-stage and advanced-stage, respectively). Sensitivity was calculated against paired PNAclamp tumor biopsy.

**Figure 3 cancers-12-03579-f003:**
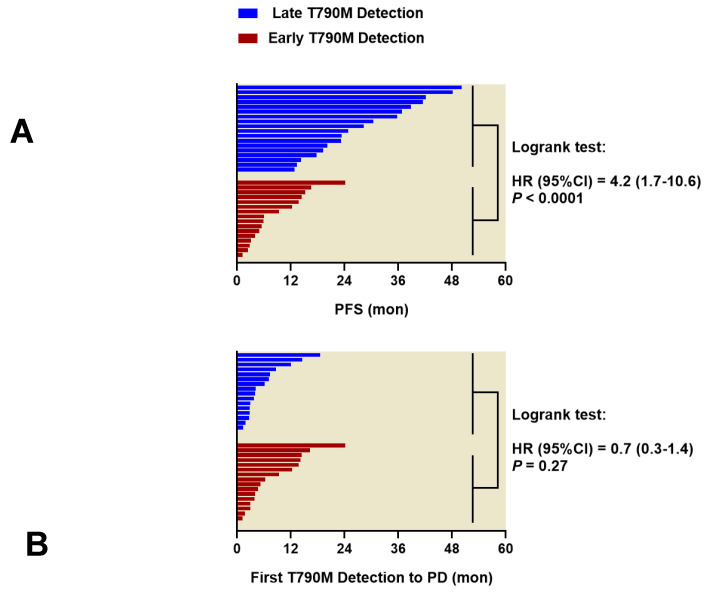
In patients treated by first-line EGFR-TKIs, (**A**) progression-free survival (PFS, interval between treatment initiation and disease progression) was significantly longer in the late T790M detection group (n = 18; first T790M detection: >3 months after treatment initiation) than in the early T790M detection group (n = 16; first T790M detection: at treatment initiation ± 3 months); (**B**) however, the interval between first T790M detection and disease progression was not significantly different between the two groups. *p*-values were obtained by Logrank test.

**Table 1 cancers-12-03579-t001:** Diagnostic performance of Sel-Cap and conventional NGS liquid biopsies for two *EGFR* activating mutations in NSCLC patients (n = 61), with paired PNAclamp tumor biopsy as the reference.

		Ex19del						L858R					
		Sel-Cap			NGS			Sel-Cap			NGS		
		Mutant	Wild	Total	Mutant	Wild	Total	Mutant	Wild	Total	Mutant	Wild	Total
**PNAclamp**	**Mutant**	15	5	20	3	15	18	11	6	17	2	14	16
	**Wild**	2	39	41	1	42	43	0	44	44	0	45	45
	**Total**	17	44	61	4	57	61	11	50	61	2	59	61
**Sensitivity**	**(95% CI)**	75%	(53–89%)		17%	(6–39%)		65%	(41–83%)		13%	(3–36%)	
**Specificity**	**(95% CI)**	95%	(84–99%)		98%	(88–100%)		100%	(92–100%)		100%	(92–100%)	
**Accuracy**	**(95% CI)**	89%	(78–94%)		74%	(62–83%)		90%	(80–95%)		77%	(65–86%)	
**Kappa**	**(95% CI)**	0.73	(0.54–0.92)		0.19	(−0.16–0.53)		0.73	(0.52–0.93)		0.17	(−0.21–0.55)	

95% CI, 95% confident interval.

**Table 2 cancers-12-03579-t002:** Diagnostic performance of Sel-Cap liquid biopsy for three *EGFR* mutations in NSCLC patients (n = 185), regardless of disease stage, with PNAclamp tumor biopsy as the reference.

Sel-Cap		Ex19del			L858R			T790M ^a^		
		Mutant	Wild	Total	Mutant	Wild	Total	Mutant	Wild	Total
**PNAclamp**	**Mutant**	50	19	69	24	12	36	7	14	21
	**Wild**	1	115	116	3	146	149	1	63	64
	**Total**	51	134	185	27	158	185	8	77	85
**Sensitivity**	**(95% CI)**	72%	(61–82%)		67%	(50–80%)		88%	(53–98%)	
**Specificity**	**(95% CI)**	99%	(95–100%)		98%	(94–99%)		82%	(72–89%)	
**Accuracy**	**(95% CI)**	89%	(84–93%)		92%	(87–95%)		82%	(73–89%)	
**Kappa**	**(95% CI)**	0.76	(0.65–0.86)		0.71	(0.57–0.85)		0.40	(0.13–0.68)	

**^a^** Only patients with time interval between tumor and plasma sample collection < 30 days (because T790M is an acquired mutation) and patients who were previously tested by PNAclamp tumor biopsy for T790M were included for data analyses (n = 85).

**Table 3 cancers-12-03579-t003:** Concordance between two NGS-based lipid biopsy platforms for *EGFR* mutations in NSCLC patients (n = 21), post development of resistance to first-line EGFR-TKIs.

No.	EGFR-TKI	Tumor	Plasma	
		PNAclamp	NGS Cancer Panel	Sel-Cap
1	Erlotinib	Ex19del ***^a^***	Ex19del	Ex19del, T790M ***^d^***
2	Erlotinib	Ex19del ***^a^***	Wild	T790M ***^d^***
3	Erlotinib	L858R ***^a^***	L858R, T790M ***^d^***	Wild
4	Erlotinib	Ex19del ***^a^***	Wild	T790M ***^d^***
5	Erlotinib	Ex19del ***^a^***	Ex19del ***^d^***	Wild
6	Gefitinib	Ex19del, T790M ***^a^***	Wild	Wild
7	Gefitinib	Ex19del ***^a^***	Ex19del, T790M	Ex19del, T790M
8	Afatinib	Ex19del ***^a^***	Wild	Wild
9	Afatinib	Ex19del ***^a^***	Ex19del	Ex19del, T790M ***^d^***
10	Gefitinib	L858R, T790M ***^b^***	L858R, T790M ***^d^***	L858R
11	Gefitinib	Ex19del, T790M ***^b^***	Wild	Wild
12	Afatinib	Ex19del ***^b^***	Ex19del	Ex19del, T790M ***^d^***
13	Erlotinib	T790M ***^c^***	Wild	Wild
14	Erlotinib	T790M ***^c^***	Wild	Wild
15	Erlotinib	L858R, T790M ***^c^***	L858R	L858R, T790M ***^d^***
16	Erlotinib	Ex19del, T790M ***^c^***	Ex19del	Ex19del
17	Erlotinib	Ex19del, T790M ***^c^***	Wild	Ex19del, T790M ***^d^***
18	Erlotinib	Ex19del, T790M ***^c^***	Ex19del	Ex19del, T790M ***^d^***
19	Gefitinib	Ex19del, T790M ***^c^***	Wild	Ex19del, T790M ***^d^***
20	Gefinitib	Ex19del, T790M ***^c^***	Wild	Ex19del, T790M ***^d^***
21	Gefinitib	L858R, T790M ***^c^***	L858R, T790M	L858R, T790M

***^a^*** No.1–9: tumor samples collected before first-line EGFR-TKI initiation. ***^b^*** No.10–12: tumor samples collected before PD diagnosis. ***^c^*** No.13–21: tumor samples collected before second-line EGFR-TKI osimertinib initiation. ***^d^*** Underlined text indicates disconcordant results between Sel-Cap and NGS cancer panel.

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
