# Peer review of "A Highly Sensitive Next-Generation Sequencing-Based Genotyping Platform for EGFR Mutations in Plasma from Non-Small Cell Lung Cancer Patients"

_cancers, 2020, doi:10.3390/cancers12123579_

Round 1
Reviewer 1 Report
Thanks for sharing this interesting work. I have enjoyed reading it, and I do not have any suggestions for editing or citing to improve the paper. I would just like to ask two questions:
- Do the authors plan any prospective validation of the technique? If so, what is the periodicity for taking and analyzing samples2.
- Do you consider that the best strategy for the detection of T790M after clinical/radiological progression to TKIs is to perform Sel-CAp and if negative tissue biopsy?
Thank you very much
Author Response
Dear reviewer
Thank you very much for your positive feedback and valuable comments, we have addressed your questions in the attached file.
Thank you for your time to help us improve the manuscript.
Best regards
Jin Hyoung Kang

Reviewer 2 Report
In this work, Shin et al. explored the use of a new digital enrichment next-generation sequencing (NGS)-based cancer panel, called Sel-Cap, for the search of Ex19del, L858R, and T790M mutations of epidermal growth factor receptor 1 (EGFR) gene in plasma samples of patients with non-small cell lung cancer (NSCLC), comparing the obtained results with those of PNAclamp tumor biopsy (reference method) and two other NGS liquid biopsies. The performed analyses led to the conclusion that Sel-Cap is a highly sensitive platform for EGFR mutations detection in plasma even in early-stage disease. Indeed, its use in a retrospective, longitudinal monitoring study highlights that early detection of the mutation T790M may allow for early prediction of disease progression.
In my opinion this study is well-written and interesting and deserves publication in Cancers.
However, some parts of the manuscript should be clarified.
I list my specific comments below.
- Abstract, line 4: ‘PNAclamp’ acronym should be specified.
- Abstract, line 7: “…following first-line EGFR-TKIs was evaluated in 34 patients, …” I should end the sentence here, inserting a period.
- Abstract, lines 9-10: “the first instance of T790M detection by a 9 sensitive liquid biopsy is critical for prediction of disease progression and guiding treatment.” Please clarify and rewrite this sentence.
- Lines 26-28: Sel-Cap approach is introduced here. Where does come from the data regarding the limit of detection? Could you provide a reference?
- Lines 29-30: “..in plasma-derived cfDNA and comparing it to..” I would eliminate ‘and’.
- Lines 31: “..and for the prediction..” I would write ‘to predict’.
- Lines 52-53: Here, I think it is appropriate to mention the principle of technique and parameters such as LOD and cut-off.
- Lines 55-56: What is the cut-off value of conventional NGS?
- Line 58: Would be appropriate better specify if the analysed loci were only those of Ex19del, L858R, and T790M mutations.
- Lines 96-97: “..and in the third study, plasma samples were 96 collected from 21 patients after they had developed resistance to first-line EGFR-TKIs and genotyped using both Sel-Cap and the NGS cancer panel..” It’s not clear to me if these 21 analysed samples were included or not in the calculation of the 185. Please clarify.
- Line 110: Please specify ‘conventional’.
- Line 129: “…T790M-positive plasma samples (n=12)..” I count 13….
- Lines 129-130: “two plasma samples detected by both tests are included in these numbers.” This sentence is not clear to me.
- Table 3: Why is first line underlined and bold? Please specify the meaning of the highlighted mutations in the legend.
- Line 146-147: “…The PFS following first-line EGFR-TKI treatment as it relates to the timing of first T790M detection are shown in Figure 3…” The sentence is not clear, please reformulate.
- Lines 151-153: “In addition, the median PFS post T790M detection is 3.6 ...” As stated in figure legend 3, I would specify that PFS post T790M detection is intended as the interval between first T790M detection and disease progression.
- Lines 176-179: A comment regarding the use of Sel-Cap for evaluation of tumor biopsy should be inserted also in the Results part.
- Line 192: I would write ‘in the current study’.
- Lines 195-198: In this point, the authors make an indirect comparison between Sel-Cap and cobas v2 tumor biopsy, by reason of the assumption that PNAclamp and cobas v2 tumor biopsy are reference methods; however I think that it should be clarified what is the accordance between these reference approaches.
- Line 204: ‘postivie’, please correct.
- Line 262: “..by disease stage because disease stage at time of sampling..”, I would write ‘this information’.
- Discussion: this part is a bit long and redundant in some points (lines 170-173 and 272-274). I think it is appropriate make it leaner.
Author Response
Dear reviewer
Thank you very much for your feedback and valuable comments, we have addressed your questions in the attached file.
Thank you for your time to help us improve the manuscript.
Best regards
Jin Hyoung Kang

Reviewer 3 Report
This is an interesting study in using liquid biopsy for EGFR detection in NSCLC patients. The study has direct clinical impact and application, which is important to the field. In fact, it is a hot topic of many investigators working on different platforms in the plasma detection. The authors concluded that “Sel-Cap is a highly sensitive platform for EGFR mutations in plasma, and the first instance of T790M detection by a 9 sensitive liquid biopsy is critical for prediction of disease progression and guiding treatment.” It is good for the authors to demonstrate the uniqueness of the study regarding relevant studies being published.
Specific Comments
- Sel-Cap’s capable for detecting EGFR 28 mutations (Ex19del, L858R, and T790M) in plasma-derived cfDNA, how about C797S and some recent emerging mutations in EGFR?
- How’s the sensitivity and specificity of genotyping by PNAclamp EGFR mutation detection kit ver.2?
- Double centrifuge is a better approach in obtaining cfDNA, which the centrifugation speed is 1,600 xg and then 16,000 xg (e.g. https://pubmed.ncbi.nlm.nih.gov/29868115/). Yet the authors use 1,200×g and 3,000×g instead. Regarding 3,000 xg is not very high speed; can the authors discuss a bot on this important pre-analytical issue? This will convince the readers that the plasma collection is cell-free without host cellular contamination.
- How’s the quality of the NGS library preparation and how’s the performance of NGS? E.g. the coverage, uniformity, MAF? Is the human genome reference used the updated one?
- Apart from DNA mutations, how about fusion genes and any novel gene transcripts being found?
- It is good to have a head-to-head or statistical analysis of the comparison in the performance of ctDNA and imaging techniques (such as computed tomography and magnetic resonance imaging) on the prognosis prediction
- It is a bit unclear which NGS referred to in Fig. 2. Besides, how about T790M? Also the x, y-axis may be swapped.
- The clinical data is not adequate, how’s the demographics of the subjects in the final results? How’s the TNM correlate with the NGS findings? Is there good evidence in showing early detection of Sel-Cap? How’s the findings relate to the metastatic site?
- What are the concordance rate between tissue and plasma in the EGFR detection? It is better to list out the details of the mutation genotypes.
- The authors claimed a highly sensitive NGS, it is good to compare to the conventional available platforms in the market to justify the claim.
- In term of an assay, what is the accuracy of the detection? How about the sensitivity, specificity, PPV, NPV, etc?
- Are there any limited of detection (LOD) being done? How little of DNA can be confidentially detected with known mutations?
- What is the turnaround time of the assay?
- Did the authors attempted and compared tumor-derived DNA from plasma to pleural effusion supernatant?
- The reference seems inadequate, some relevant studies and papers not cited. Some papers may be included to enrich the discussion context. Some examples are listed below but not exclusively.
Romero A, Jantus-Lewintre E, García-Peláez B, et al., Comprehensive cross-platform comparison of methods for non-invasive EGFR mutation testing: Results of the RING observational trial. Mol Oncol. 2020 Oct 26.
Ding PN, Becker T, Bray V, et al., Plasma next generation sequencing and droplet digital PCR-based detection of epidermal growth factor receptor (EGFR) mutations in patients with advanced lung cancer treated with subsequent-line osimertinib. Thorac Cancer. 2019;10(10):1879-1884.
Xue VW, Wong CSC, Cho WCS. Early detection and monitoring of cancer in liquid biopsy: advances and challenges. Expert Rev Mol Diagn. 2019;19(4):273-276.
Zugazagoitia J, Gómez-Rueda A, Jantus-Lewintre E, et al., Clinical utility of plasma-based digital next-generation sequencing in oncogene-driven non-small-cell lung cancer patients with tyrosine kinase inhibitor resistance. Lung Cancer. 2019;134:72-78.
Liang W, Cai K, Chen C, et al., Society for Translational Medicine consensus on postoperative management of EGFR-mutant lung cancer (2019 edition). Transl Lung Cancer Res. 2019;8(6):1163-1173.
Garcia J, Wozny AS, Geiguer F, et al., Profiling of circulating tumor DNA in plasma of non-small cell lung cancer patients, monitoring of epidermal growth factor receptor p.T790M mutated allelic fraction using beads, emulsion, amplification, and magnetics companion assay and evaluation in future application in mimicking circulating tumor cells. Cancer Med. 2019;8(8):3685-3697.
Author Response

(The authors gave the same response as above.)

Round 2
Reviewer 3 Report
Reply: Thank you very much for your careful review. The plasma was obtained by centrifugation of 1,200g for 15 min, followed by 13,000g for 10 min. The method was adopted from a previous study (Xue VW et al., Front Genet 2018). In the submitted manuscript, 13,000g was incorrectly noted as 3,000g, we apologize for this mistake and have revised the manuscript (line 64).
In order to let the reader knows the method source/reference, the authors better cite the captioned reference in the paper.
Author Response
Dear reviewer,
Thank you very much for your comment.
We have added the reference into the method.
Please see the revised manuscript (line 66).
Best Regards,
Jin Hyoung Kang